# A randomized trial of Trendelenburg position for acute moderate ischemic stroke

Hui-Sheng Chen [1] ✉, Nan-Nan Zhang[1], Yu Cui[1], Xiao-Qiu Li[1], Cheng-Shu Zhou[2], Yu-Tong Ma[3], Hong Zhang[4], Chang-Hao Jiang[5], Run-Hui Li[6], Li-Shu Wan[7], Zhen Jiao[8], Hong-Bo Xiao[8], Zhuo Li[9], Ting-Guang Yan[10], Duo-Lao Wang[11] & Thanh N. Nguyen [12]

We aim to explore the effect of head-down position (HDP), initiated within 24 hours of onset, in moderate anterior circulation stroke patients with probable large artery atherosclerosis (LAA) etiology. This investigator-initiated, multi-center trial prospective, randomized, open-label, blinded-endpoint, multi-center and phase-2 trial was conducted in China and completed in 2021. Eligible patients were randomly assigned (1:1) into the HDP group receiving −20° Trendelenburg, or control group receiving standard care according to national guideline. The primary endpoint was proportion of modified Rankin Scale (mRS) of 0 to 2 at 90 days, which is a scale for measuring the degree of disability after stroke. 90-day mRS was assessed by a certified staff member who was blinded to group assignment. A total of 96 patients were randomized (47 in HDP group and 49 in control group) and 94 (97.9%) patients were included in the final analysis: 46 in HDP group and 48 in control group. The proportion of favorable outcome was 65.2% (30/46) in the HDP group versus 50.0% (24/48) in the control group (unadjusted: OR 2.05 [95%CI 0.87-4.82], $P = 0.099$). No severe adverse event was attributed to HDP procedures. This work suggests that the head-down position seems safe and feasible, but does not improve favorable functional outcome in acute moderate stroke patients with LAA. This trial was registered with ClinicalTrials.gov, NCT03744533.

To date, there is a paucity of effective neuroprotective treatments for acute ischemic stroke (AIS), other than reperfusion therapy such as intravenous thrombolysis and mechanical thrombectomy, which is limited by a strict therapeutic time window and requirement for a highly developed stroke system of care[1]. The effect of head position (lying-flat vs sitting-up position) as a nonpharmacological therapy on stroke has been investigated[2–5], but the inconsistent results have led to current ambiguous guideline recommendations[1–6]. It is generally accepted that the supine position may increase blood flow and improve oxygenation[2–4,7–9], but with potential risks such as increased intracranial pressure, cardiopulmonary dysfunction, and aspiration pneumonia[10–13]. The neutral results of the Head Positioning in Acute

[1]Department of Neurology, General Hospital of Northern Theater Command, Shenyang 110016, China. [2]Department of Neurology, Anshan Changda Hospital, Anshan 114000, China. [3]Department of Neurology, Beipiao Central Hospital, Beipiao 122100, China. [4]Department of Neurology, Fukuang General Hospital of Liaoning Health Industry Group, Fushun 113005, China. [5]Department of Neurology, The Traditional Medicine Hospital of Dalian Lvshunkou, Dalian 116045, China. [6]Department of Neurology, Central Hospital affiliated to Shenyang Medical College, Shenyang 110024, China. [7]Department of Neurology, Dandong First Hospital, Dandong 118015, China. [8]Department of Neurology, Anshan Central Hospital, Anshan 114000, China. [9]Department of Neurology, Panjin Central Hospital, Panjin 124010, China. [10]Department of Neurology, Chaoyang Central Hospital, Chaoyang 122099, China. [11]Department of Clinical Sciences, Liverpool School of Tropical Medicine, Liverpool, UK. [12]Neurology, Radiology, Boston Medical Center, Boston, MA, USA. ✉e-mail: chszh@aliyun.com

Stroke Trial (HeadPoST) may have been due to the broad inclusion of stroke patients, particularly of patients with milder deficits, which was a key criticism of the trial. While patients with large artery atherosclerosis (LAA) etiology could be a suitable target population[14], subgroup analysis of HeadPoST did not detect any evidence of heterogeneity of treatment effect across clinician-diagnosed stroke subtypes[15].

In theory, compared with the supine or lying-flat position, the steeper head-down position (i.e., fully supine with Trendelenburg[16]) could significantly increase blood flow to the ischemic penumbra and improve oxygenation of the brain in the first hours or days after stroke[17]. Our recent experiment in a rat animal model with middle cerebral artery occlusion showed that the head-down position (HDP) with −30° and 2 h duration after ischemia could improve neurological function and reduce infarct volume[18]. Moreover, we anecdotally observed several LAA patients in our center, after which a HDP (−20°) averted neurological deterioration and improved clinical outcomes[19].

In this work, we undertook the prospective, multicenter, randomized, open-label, blinded-endpoint trial to explore the effect of HDP, initiated within 24 h of symptom onset, in moderate AIS patients with LAA who were not eligible for intravenous thrombolysis or endovascular therapy.

## Results
### Trial population
Between Nov 16, 2018, and Aug 28, 2021, 113 consecutive patients were screened and 96 eligible patients were randomly assigned to the HDP group (n = 47) and control group (n = 49). After two patients were excluded, 94 patients were included in the mITT population (46 in the HDP group and 48 in the control group, Fig. 1). The procedure was completed according to the protocol for 89 patients (42 in the HDP group and 47 in the control group), which was included in the per-protocol analysis (Fig. 1). There were no cross-overs between groups in the trial. No patient received carotid or intracranial revascularization. Enrollment was completed in May 2021.

Baseline characteristics were well balanced between groups in the mITT population (Table 1) and in the per-protocol population (Supplementary Table 1). HDP procedure details were shown in Table 2. The median duration of position intervention within 24 h was 15.0 h (IQR 12.0–16.0), and the median duration of each intermittent HDP was 35.0 minutes (IQR 30.0–60.0). During the treatments, the side-lying position was used in 26.1% (12/46) patients to prevent possible aspiration.

### Primary and secondary outcomes
For the primary outcome, the proportion of mRS score 0–2 at 90 days was 65.2% (30/46) in the HDP group and 50.0% (24/48) in the control group (unadjusted OR 2.05 [95% CI 0.87–4.82], p = 0.099; Table 3 and Fig. 2). Similar OR results were observed in the per-protocol analysis (Supplementary Table 2), in the last observation carried forward, worst-case scenario, and best-case scenario sensitivity analyses (Supplementary Table 3), and after adjustment for the prespecified prognostic variables (Table 2).

For the secondary outcome, the proportion of mRS score 0–1 at 90 days was 45.7% (21/46) in HDP group and 25.0% (12/48) in the control group (unadjusted OR 2.66 [95% CI 1.10–6.44], p = 0.030; Table 3 and Fig. 2). There was a significant difference between the two groups based on 90-day ordinal shift analysis (unadjusted OR 2.70 [95% CI 1.27–5.72], p = 0.010, Table 3) in favor of HDP. There was also a significant difference in NIHSS change from baseline to day 12 after randomization between the two groups (3.3 [3.3] vs 0.6 [6.9], unadjusted OR −0.15 [95% CI −0.25 to −0.05], p = 0.004, Table 3). In the per-protocol analysis, significant differences in odds of having an mRS score 0 to 1, mRS improvement within 90 days, and NIHSS change from baseline to day 12 after randomization were also found between groups in both unadjusted and adjusted analysis (Supplementary Table 2). END occurred in 4.2% (2/48) of patients in the control group. There were 6 (6.4%) of 94 patients who died during the follow-up period, including 2.2% (1/46) in the HDP group and 10.4% (5/48) in the control group (unadjusted OR 0.20 [95% CI 0.02–1.74], p = 0.144, Table 3).

### Safety and adverse events
Adverse events are reported in Table 4. There were eight HDP-related adverse events: five patients reported headaches, two developed anxiety, and one experienced fear, but these AEs resolved after adjusting the patient to the horizontal position without any further medical treatment. There was no difference in asymptomatic

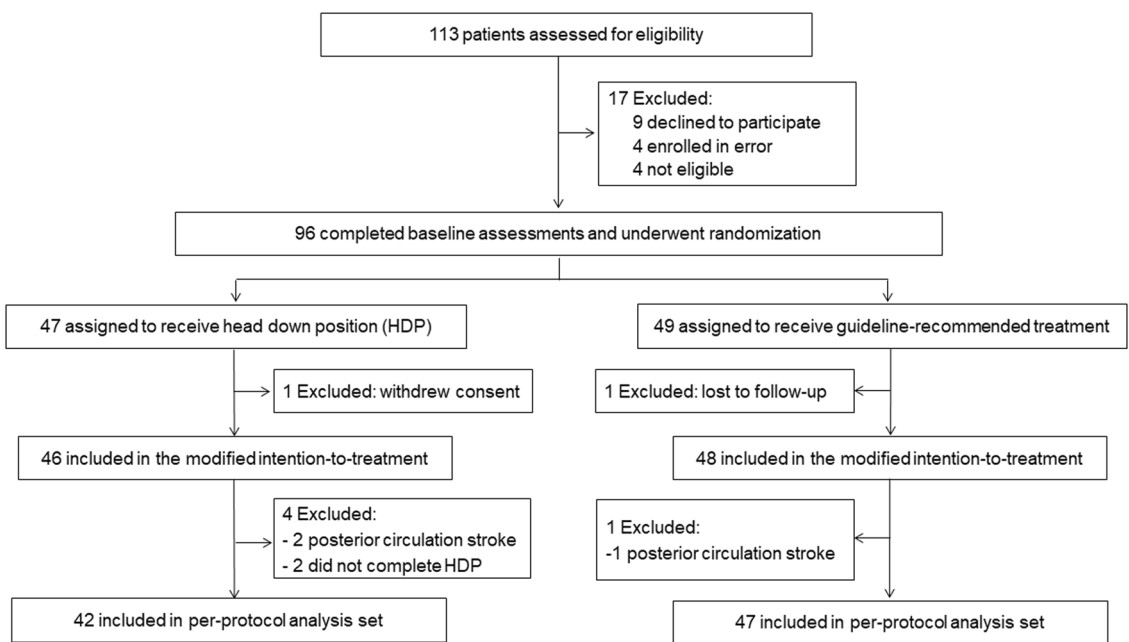

**Fig. 1 | Trial profile.** This figure shows the overall patient flow in the trial. HDP head-down position.

**Table 1 | Baseline characteristics and procedural details for the modified intention-to-treat population**

| | HDP group (n = 46) | Control group (n = 48) | P value |
|---|---|---|---|
| Age, years | 60.9 (11.2) | 64.2 (11.7) | 0.172 |
| Sex | | | 0.898 |
| Male | 33 (71.7%) | 35 (72.9%) | |
| Female | 13 (28.3%) | 13 (27.1%) | |
| Risk factors | | | |
| Hypertension | 26/45 (57.8%) | 30/47 (63.8%) | 0.552 |
| Diabetes | 14/45 (31.1%) | 12/47 (25.5%) | 0.552 |
| Hyperlipidemia | 3 (6.5%) | 3/47 (6.4%) | 0.978 |
| Coronary heart disease | 7 (15.2%) | 3/47 (6.4%) | 0.169 |
| Previous stroke | 16 (34.8%) | 16 (33.3%) | 0.882 |
| Current smoker | 26 (56.5%) | 25 (52.1%) | 0.666 |
| Current drinker | 24 (52.2%) | 21 (43.8%) | 0.414 |
| Blood pressure at randomization, mmHg | | | |
| Systolic | 161.5 (25.5) | 159.3 (24.8) | 0.674 |
| Diastolic | 89.5 (13.9) | 88.0 (11.5) | 0.590 |
| Blood pressure at 24 h, mmHg | | | |
| Systolic | 134.3 (60.0) | 126.7 (62.8) | 0.564 |
| Diastolic | 94.9 (28.9) | 99.6 (31.6) | 0.475 |
| Blood pressure at 7 days, mmHg | | | |
| Systolic | 133.0 (29.1) | 130.6 (27.6) | 0.718 |
| Diastolic | 92.8 (28.4) | 88.7 (24.3) | 0.529 |
| NIHSS score at randomization[a] | 9 (7–10) | 9 (6–11) | 0.742 |
| Antithrombotic therapy | | | 0.956 |
| Mono antiplatelet | 19 (41.3%) | 20/47 (42.6%) | |
| Dual antiplatelet | 16 (34.8%) | 15/47 (31.9%) | |
| Antiplatelet + anticoagulant | 11 (23.9%) | 12/47 (25.5%) | |
| Lipid-lowering therapy | | | 0.230 |
| High intensity | 30/42 (71.4%) | 26/44 (59.1%) | |
| Nonhigh intensity | 12/42 (28.6%) | 18/44 (40.9%) | |
| Responsible vessels | | | 0.335 |
| Extracranial ICA | 6/43 (14.0%) | 4/46 (8.7%) | |
| Intracranial ICA | 9/43 (20.9%) | 5/46 (10.9%) | |
| M1 segment of MCA | 26/43 (60.5%) | 36/46 (78.3%) | |
| The vertebral or basilar artery | 2/43 (4.7%) | 1/46 (2.2%) | |
| Degree of responsible vessel stenosis | | | 0.979 |
| Moderate (50–69%) | 9/43 (20.9%) | 9/45 (20.0%) | |
| Severe (70–99%) | 12/43 (27.9%) | 12/45 (26.7%) | |
| Occlusion | 22/43 (51.2%) | 24/45 (53.3%) | |
| Onset to randomization time (h) | 14.5 (6.8-21.0) | 10.0 (7.0-18.5) | 0.242 |
| ICU care | 11 (23.9%) | 8 (51.1%) | 0.382 |

Data are No.(%) or No./total (%), mean (SD), or median (IQR). Baseline characteristics were compared with Student's *t*-test if normally distributed or Mann–Whitney test if not normally distributed for continuous variables, and $\chi^2$ for categorical variables. All tests were two-tailed. *HDP* head-down position, *ICA* internal carotid artery, *MCA* middle cerebral artery, *HDP* head-down position, *NIHSS* National Institute of Health Stroke Scale.
[a]Scores range from 0 to 42, with higher scores indicating a more severe neurological deficit.

**Table 2 | HDP procedure in the modified intention-to-treat population**

| Duration within first 24 h, h, median (IQR) | 15.0 (12.0–16.0) | |
|---|---|---|
| Average duration of each HDP after 24 h (min) | 35.0 (30.0–60.0) | |
| Number of side-lying HDP | 12 (26.1%) | |
| Frequency each day | | |
| Days of intermittent HDP, no. (%) | | |
| | 3 times | ≤2 times |
| ≤8 | 3/37 (8.1%) | 1/9 (11.1%) |
| 9 | 7/37 (18.9%) | 2/9 (22.2%) |
| 10 | 1/37 (2.7%) | 0 |
| 11 | 10/37 (27.0%) | 1/9 (11.1%) |
| 12 | 3/37 (8.1%) | 1/9 (11.1%) |
| 13 | 6/37 (16.2%) | 1/9 (11.1%) |
| 14 | 7/37 (18.9%) | 3/9 (33.3%) |

Data were no.(%) or no./total (%).
*HDP* head-down position.

intracranial hemorrhage, the occurrence of stroke and cardiovascular events, and stroke-associated pneumonia between the two groups.

## Discussion

In this investigator-initiated, randomized, multicenter trial, we investigated the effect of HDP (−20°) on functional outcomes in acute moderate ischemic stroke patients with probable LAA. We found that treatment with HDP for 2 weeks, applied as an adjunct to guideline-based medical management, was safe and did not improve the primary outcome (mRS 0–2 at 90 days), when compared with guideline-based medical management alone. However, HDP may have an improved effect on major secondary outcomes, including excellent functional outcome (mRS 0–1), ordinal shift distribution of mRS at 90 days, END, and change in NIHSS.

Many studies have investigated the effect of head position on cerebral blood flow and cerebral perfusion[3,4,8,9,20], and HeadPoST was the first large study to compare the effect of supine versus sitting position on neurological improvement in AIS patients. However, the HeadPoST trial did not find the effect of different head positions on long-term neurological outcomes[5]. Furthermore, the post hoc analyses of the HeadPoST study of patients with moderate–severe AIS showed that the flat head position early after stroke symptom onset was not associated with functional recovery, but could be effective in patients with low NIHSS and large vessel occlusion[21].

There are several differences between the current study and prior studies. First, the head position at −20° was adopted in the current study, while a horizontal supine versus sitting position was utilized in prior studies[2-5,8,9,20]. In theory, HDP can enhance cerebral blood flow and increase cerebral perfusion compared with a supine or sitting position due to the force of gravity[22]. The proposal was also supported by our recent data that HDP of −20° can improve neurological function, reduce brain edema and infarct volume in rats with middle cerebral artery occlusion/reperfusion model[18] and prevent neurological deterioration in several AIS patients with LAA[19]. Secondly, there was a difference in enrolled participants: the current study enrolled acute moderate ischemic stroke patients with probable LAA, while AIS patients in the HeadPoST study did not classify the stroke severity or stroke etiology. We contended that stroke patients with moderate neurological deficits would be the target population most likely to benefit from neuroprotective therapy, because the neuroprotective effect could be underestimated in patients with mild neurological deficits, whereas patients with severe neurological deficits due to large artery occlusion would be less likely to benefit from neuroprotective treatment without reperfusion therapy[1,23]. We chose the LAA stroke subtype as the target, because the mechanism of stroke and recurrent stroke in these patients is often related to hypoperfusion[24], whose neurological function may be improved from increased cerebral perfusion and recruitment of the collateral circulation[25,26] due to a head-down position. Optimizing cerebral perfusion is critical for the treatment of ischemic stroke patients, especially those with LAA who rely on collateral circulation[17,19,20,27,28]. In this trial, ~80% of enrolled patients harbored severe stenosis or vessel occlusion. This subgroup of

**Table 3 | Primary and secondary outcomes in the modified intention-to-treat population**

| | HDP group (n = 46) | Control group (n = 48) | Unadjusted | | Adjusted[a] | |
|---|---|---|---|---|---|---|
| | | | OR (95% CI) | P value | OR (95% CI) | P value |
| Primary outcome | | | | | | |
| mRS score 0–2 at 90 days | 30 (65.2%) | 24 (50.0%) | 2.05 (0.87–4.82) | 0.099 | 2.28 (0.84–6.14) | 0.104 |
| Secondary outcomes | | | | | | |
| mRS score 0–1 at 90 days | 21 (45.7%) | 12 (25.0%) | 2.66 (1.10–6.44) | 0.030* | 2.72 (1.00–7.36) | 0.049* |
| Improvement in mRS according to category at day 90[b] | | | 2.70 (1.27–5.72) | 0.010* | 3.15 (1.44–6.92) | 0.004* |
| 0 | 11 (23.9%) | 3 (6.3%) | | | | |
| 1 | 10 (21.7%) | 9 (18.8%) | | | | |
| 2 | 9 (19.6%) | 12 (25.0%) | | | | |
| 3 | 10 (21.7%) | 11 (22.9%) | | | | |
| 4 | 1 (2.2%) | 6 (12.5%) | | | | |
| 5 | 2 (4.3%) | 1 (2.1%) | | | | |
| 6 | 1 (2.2%) | 5 (10.4%) | | | | |
| Early neurological deterioration within 48 h[c] | 0 | 2 (4.2%) | | | | |
| Change in NIHSS score at day 12 from baseline[d] | 3.3 (3.3) | 0.6 (6.9) | −0.15 (−0.25–−0.05) | 0.004* | −0.16 (−0.26–−0.06) | 0.002* |
| Death within 90 days | 1 (2.2%) | 5 (10.4%) | 0.20 (0.02–1.74) | 0.144 | 0.26 (0.02–3.60) | 0.312 |

Frequency data were no.(%) or mean (SD). The treatment effect is presented as the odds ratio (95% CI) of the HDP group versus the control group, analyzed by unadjusted and adjusted binary logistic regression. The treatment effect is presented as a geometric mean ratio. All tests were two-tailed. No adjustments were made for multiple comparisons. Details be provided as the method (in-person vs telephone) and source (patient vs surrogate) of the mRS outcomes.

*HDP* head-down position, *mRS* modified Rankin scale, *NIHSS* National Institute of Health Stroke Scale.

[a]Adjusted for key prognostic covariates (age, NIHSS score at randomization, the degree of responsible vessel stenosis, onset to randomization time, and location of responsible vessels).

[b]The outcome was an assessment of scores across all seven levels of the mRS (ranging from 0 [no symptoms] to 6 [death]), done using a shift analysis of the ordinal data.

[c]Early neurological deterioration was defined as ≥4 increase in NIHSS score within 48 h, but not the result of a cerebral hemorrhage.

[d]NIHSS scores range from 0 to 42, with higher scores indicating greater stroke severity. Log (NIHSS + 1) was analyzed using a generalized linear model.

*P < 0.05.

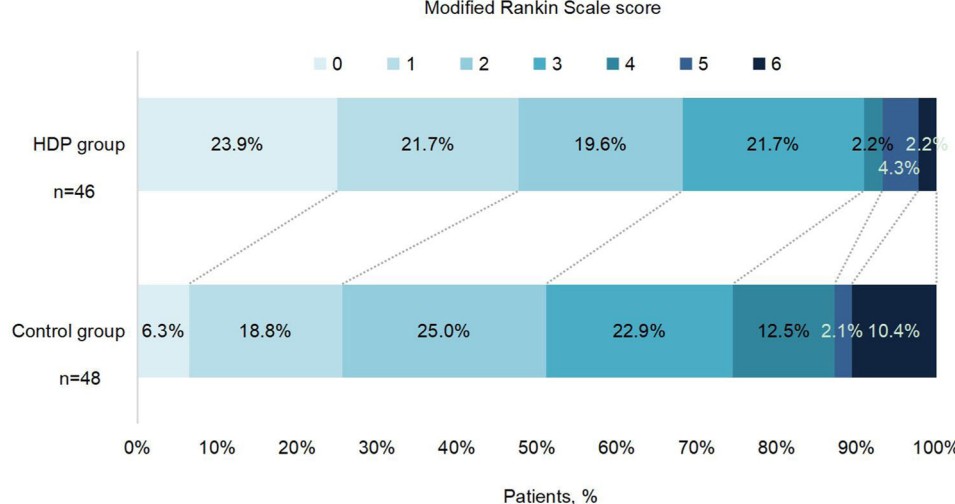

**Fig. 2 | Distribution of modified Rankin scale scores at 90 days by treatment groups in the modified intention-to-treat population.** Raw distribution of scores is shown. Scores range from 0 to 6: 0 = no symptoms, 1 = symptoms without clinically significant disability, 2 = slight disability, 3 = moderate disability, 4 = moderately severe disability, 5 = severe disability, and 6 = death. HDP head-down position. Source data are provided as a Source Data file.

patients is most vulnerable to stroke progression or recurrence associated with hypoperfusion mechanisms, and hence may derive greater benefit from the head-down position.

Third, we implemented different head position interventions over time: 15 h duration within 24 h of presentation, followed by three times daily for 2 weeks in the current study, while only the first 24 h after randomization were implemented in the HeadPoST study. We postulate that longer-term HDP intervention may result in more benefits of improved cerebral perfusion, given that there may be the presence of long-lasting penumbra in this population[29]. Given the possible increase in intracranial pressure due to HDP, the suitable HDP strategy, such as

the angle and duration time warrant further investigation. Altogether, the positive direction of the effect of HDP on AIS in the current study could be attributed to the targeted population (moderate neurological deficits, median NIHSS score 9 [7–10], with LAA etiology), the lower head position, and the longer duration of head position (10 to 14 days vs 1 day), possibly through increasing cerebral blood flow including collateral circulation improvement by gravitational force[22,25].

For secondary outcomes, we found a significant mRS improvement at 90 days in the HDP group vs the control group, as well as improvement in NIHSS from baseline to day 12. We also observed a significant improvement in the proportion of mRS scores 0–1 at

**Table 4 | Adverse events**

| | HDP group (n = 46) | Control group (n = 49) | P value |
|---|---|---|---|
| Stroke or other vascular events within 90 days | | | |
| Asymptomatic intracranial hemorrhage[a] | 1 (2.2%) | 1 (2.0%)[b] | 0.96 |
| Recurrence of ischemic stroke[a] | 1/44 (2.3%) | 4/47 (8.5%) | 0.22 |
| Cardiovascular event[a] | 0 | 1 (2.0%)[b] | 1.00 |
| Stroke-associated pneumonia[a] | 0 | 1 (2.0%)[b] | 1.00 |
| The number of patients with HDP-related adverse events[c] | | | |
| Headache | 5 (10.9%) | NA | |
| Anxiety | 2 (4.3%) | NA | |
| Fear | 1 (2.2%) | NA | |

Frequency data were no.(%) or no./total (%).
HDP head-down position, NA not applicable.
[a]Serious adverse events.
[b]One patient had intracranial hemorrhage, a cardiovascular event, and stroke-associated pneumonia at the same time.
[c]The adverse events were not present at the beginning of the study, and whether the unexpected adverse events were associated with HDP will be further adjudicated by the principal investigator. All tests were two-tailed. No adjustments were made for multiple comparisons.

90 days in the HDP group. For safety outcomes, there was no difference in mortality between the two groups, and no END in the HDP group. Collectively, these results support the safety and potentially improved neurological outcomes with HDP in patients with acute moderate ischemic stroke with LAA in a Chinese population.

In our study, other safety endpoints, including pneumonia and cardiovascular events, were similar between the two groups. Pneumonia is a major risk factor for death after acute stroke. Preventive administration of antibiotics is superior in reducing infections after severe ischemic stroke[30]. The effect of head position on aspiration pneumonia is a common concern[12,31]. In our study, patients with a high risk of aspiration pneumonia were allowed to lie in a head-down position at least half an hour after their meal, and a prone or side-lying position was recommended. Another safety concern that has been raised is the effect of a head-down position on cardiac function[13,32]. All patients in the head-down group received electrocardiogram monitoring during the intervention, and no cardiac-related adverse event was found. Furthermore, no difference in brain natriuretic peptide (BNP) change was found between the two groups (Supplementary Table 4). In the HDP group, only a few patients reported uncomfortable symptoms, such as headache, anxiety, and fear, which resolved after adjusting to a horizontal position without any medical treatment.

The strength of this study is the randomized multicenter design to determine the safety and possible efficacy of HDP in acute moderate ischemic stroke patients with probable LAA etiology. These promising results may promote further trials to investigate the effect of HDP in a broader array of AIS patients, for example, patients who develop early neurological deterioration due to hypoperfusion mechanisms, in addition to the current patients. We acknowledge several limitations to our study. The main limitation is the relatively small sample due to the pilot nature, which makes the conclusion exploratory and subgroup analysis, such as the effect of site on primary outcome, impossible. Second, the open-label design may have resulted in bias, although we used blinded evaluation at 90 days to mitigate this potential bias. Third, the highly selected population, for example, excluding patients who received thrombolysis or thrombectomy, limited to anterior circulation stroke, introduce selection bias and may limit the generalizability of our results. Fourth, the neuroimaging infarct or penumbra size was not determined in detail in the pilot study. Finally, there was no limit of head position in the control group in this trial, but we did not record the actual head position in the control group during the

trial, especially within 24 h after randomization. This detailed information would be important to understand the effect of different head positions on stroke outcomes.

In conclusion, this randomized clinical trial suggests that in patients with acute moderate ischemic stroke with large artery atherosclerosis, the head-down position seems safe and feasible, but does not improve 90-day favorable functional outcome as a primary outcome, although a direction of benefit was present with the potential to improve secondary outcomes. A prospective, large-sample, multicenter trial is warranted to confirm these findings.

## Methods
### Study design
HOPES2 (Head-dOwn Position for acutE moderate ischemic Stroke with large artery atherosclerosis) was an investigator-initiated, prospective, randomized, open-label, blinded-endpoint (PROBE), multicenter and phase-2 trial to assess the feasibility, safety and possible efficacy of two weeks of HDP in moderate AIS-LAA patients within 24 h from symptom onset. Due to the small sample size and no involving genetic information and materials, this trial was waived approval from China's Ministry of Science and Technology related to the export of genetic information and materials. The trial was conducted at 10 medical sites (Supplementary Note 7 in Supplementary information) in China, and approved by the ethics committees of the General Hospital of Northern Theater Command (former General Hospital of Shenyang Military Region, IRB: k (2018)38), Anshan Changda Hospital, Beipiao Central Hospital, Fukuang General Hospital of Liaoning Health Industry Group, the Traditional Medicine Hospital of Dalian Lvshunkou, Central Hospital affiliated to Shenyang Medical College, Dandong First Hospital, Anshan Central Hospital, Panjin Central Hospital, and Chaoyang Central Hospital. Signed informed consents were obtained from the patients or their legally authorized representatives.

### Participants
Eligible patients were adults aged 18 years or older with acute moderate ischemic stroke (defined as baseline National Institutes of Health Stroke Scale [NIHSS] scores 6 to 16) with probable LAA etiology at the time of randomization who had been functioning independently in the community (modified Rankin Scale [mRS] scores 0 to 1; range 0 [no symptoms] to 6 [death]) before the stroke, and were enrolled up to 24 hours after onset of stroke symptoms (defined as the time the patient was last seen well). Head and neck CTA or MRA imaging were done on admission to identify AIS patients with probable LAA etiology based on the Trial of Org 10172 in Acute Stroke Treatment (TOAST) criteria[27] (responsible artery ≥50% stenosis or occlusion, confirmed by CTA or MRA). Key exclusion criteria were that a patient received intravenous thrombolysis and/or endovascular therapy, other etiologies, such as cardiogenic embolism, arteritis, arterial dissection, moyamoya disease; planned carotid or intracranial revascularization within 90 days, any possible contraindication to head-down position (e.g., active vomiting, pneumonia, uncontrolled heart failure, and need for enteral feedings), and patients with neurological fluctuations whose neurological deficits are not eligible within 24 h after symptom onset. A full list of inclusion and exclusion criteria is available in the study protocol (Supplementary Notes 1–3 in Supplementary information).

### Randomization and masking
In this trial, eligible patients were randomly assigned (1:1) using a computer-generated randomization sequence with a block size of four and sealed envelopes, prepared by an independent statistician, into either HDP group receiving Trendelenburg as an adjunct to guideline-based medical management, or a control group only receiving guideline-based medical management. The final 90-day mRS was evaluated by one qualified personnel who was blinded to treatment allocation according to a standardized procedure manual in each study center. Central

adjudication of clinical and safety outcomes was also conducted by assessors unaware of treatment allocation or clinical details.

## Procedures
In the HDP group, patients were positioned to −20° Trendelenburg position from 8:00 a.m. to 10:00 p.m. within the first 24 h after randomization. During this period, the patients were continuously monitored by ECG and blood oxygen saturation, and asked to report any discomfort. If the patients could not tolerate this position, they were then adjusted slowly to a horizontal position for 10 to 30 min, and then returned to −20°. The repositioning could be repeated during HDP treatment. After 24 h, patients were placed in a −20° Trendelenburg position with 1 to 1.5 h duration three times a day, from 9:00–11:00, 15:00–17:00, and 20:00–22:00, respectively. The treatment procedure lasted for 10 to 14 days. During the treatments, the side-lying position with −20° Trendelenburg was allowed if there was a high risk of aspiration suspected by local providers. In the control group, patients were treated according to the AHA/ASA 2018 guidelines for the early management of ischemic stroke without any intervention of head position (supine or sitting position determined by local investigator).

Neurological status, measured with the NIHSS, was assessed at baseline, 7 days, and 12 days after randomization. Demographic and clinical details were obtained at randomization. Follow-up data were collected at 7 days, 12 days (or at hospital discharge if earlier), and 90 days after randomization. Remote and on-site quality control monitoring and data verification were performed throughout the study. All patients received standard medical management according to national stroke guidelines[33].

## Outcomes
In the original design, the primary endpoint was the proportion of excellent functional outcome, defined as an mRS score of 0–1 at 90 days (Supplementary Note 1 in Supplementary information). Given the relatively poor prognosis of patients with moderate AIS-LAA stroke, the primary outcome was changed to the proportion of favorable functional outcome defined as a 90-day mRS score of 0–2, after the steering committee discussion on March 19, 2019 (Supplementary Note 2 in Supplementary information), when 16 patients were enrolled. Accordingly, secondary outcomes included mRS score 0–1 at 90 days, early neurological deterioration (END), change in NIHSS score at day 12 compared with baseline, the occurrence of stroke or other vascular events, and death due to any cause within 90 days. END was defined as ≥4 point increase in NIHSS score within 48 h, but was not a result of intracerebral hemorrhage (Supplementary Methods in Supplementary information).

Prespecified safety outcomes included any adverse events and serious adverse events during HDP, such as patient fear, headache, anxiety, intracranial hemorrhage, cardiopulmonary events, and pneumonia, which were not present at the beginning of the study. Adverse events with HDP were adjudicated by the chairman of the data safety monitoring board (YLW).

## Statistical analysis
No formal sample size calculation was performed due to no relevant data available from previous trials. For this exploratory trial, the sample size (50 patients per group) was based on the recommendation of the Steering Committee. Statistical analyses were performed on a modified intention-to-treat (mITT) principle, which comprised of patients who are randomized, regardless of whether they prematurely discontinue treatment or are otherwise protocol violators/deviators. Participants who lost to follow-up or withdrew will not be included in the mITT population. Baseline characteristics and procedural details were compared with Student's $t$-test if normally distributed or Mann–Whitney test if not normally distributed for continuous variables, and $\chi^2$ for categorical variables. The treatment effect is presented as the odds ratio (95% CI) of the HDP group versus the control group, analyzed by binary logistic regression. Shift analysis of the mRS scores at 90 days was performed using ordinal logistic regression. Change in log (NIHSS score) between admission and at 12 days was compared using a generalized linear model, and the geometric mean ratios between HDP and control groups with their 95% CIs were derived. In sensitivity analyses, missing values in the primary outcome were imputed using the last observation carried forward method, the worst-case scenario, best-case scenario approaches, and the primary outcomes were adjusted for confounding covariates (age, NIHSS score at randomization, the degree of related vessel stenosis, onset to randomization time, and location of responsible vessels). The missing values of baseline variables in the covariate-adjusted analyses were imputed using simple imputation methods based on their sample distributions. Descriptive statistics of proportions were used for the adverse events data. Continuous data are presented as mean (SD) or median (IQR) as appropriate. For categorical variables, absolute and relative frequencies are presented. The alpha error level was set at 0.05. Further information about the statistical analyses plan is shown in Supplementary Notes 4–6 in Supplementary information. Analyses were done using the statistical software IBM SPSS Statistics 24.

This trial of HOPES2 was registered with ClinicalTrials.gov with the number NCT03744533 on November 16, 2018, and is now closed at all participating sites.

## Reporting summary
Further information on research design is available in the Nature Portfolio Reporting Summary linked to this article.

## Data availability
De-identified data collected for the study, including age, sex, baseline NIHSS score, treatment allocation, and functional outcome, will be shared beginning 3 months and ending 5 years following publication by requesting the corresponding author (Hui-Sheng Chen, email: chszh@aliyun.com) for academic purposes. The corresponding author will reply to the request within 2 months, subject to the approval of the ethics committees of the General Hospital of Northern Theater Command. Source data are provided with this paper.

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

## Acknowledgements

We thank the investigators and research staff of the HOPES2 trial (Supplementary Note 7 in Supplementary information) at the participating sites and members of the trial steering and data monitoring committees (Supplementary Note 7 in Supplementary information). We also thank the participants, their families, and friends. This study was funded by grants from the National Natural Science Foundation of the Peoples Republic of China (8207147) and the Science and Technology Project Plan of Liao Ning Province (2018225023 and 2019JH2/10300027).

## Author contributions

N.-N.Z. and H.-S.C. wrote the first draft of the manuscript. H.-S.C. designed the study and critically revised the manuscript. N.-N.Z., Y.C., X.-Q.L., C.-S.Z., Y.-T.M., H.Z., C.-H.J., R.-H.L., L.-S.W., Z.J., H.-B.X., Z.L., and T.-G.Y. participated in data collection. N.-N.Z., Y.C., and D.-L.W. analyzed the data. T.N.N. critically revised the manuscript. All authors vouch for the data and analysis and contributed to writing the paper.

## Competing interests

The authors declare no competing interests.
