## [Peer Review File · Nature Communications]

REVIEWER COMMENTS

Reviewer #1 (Remarks to the Author):

This is a randomized controlled trial, which investigated efficacy and safety of head-down opposition during the first 24h among 94 patients with acute ischemic stroke.

1 Sample size of 94 patients seems to be too small, and this trial seems to be underpowered to show efficacy on the primary outcome of functional status at 90 days.

2 Associated with insufficient sample size, there seems to be some imbalance in important baseline factors, such as age, diabetes and systolic BP.

3 The authors did not provide information on stroke management (ICU care, thrombolysis, endovascular treatment etc) which can affect primary outcome of physical function, although imbalance in stroke management can bias the results of this trial.

Reviewer #2 (Remarks to the Author):

There are several questions regarding the statistical analysis part:

1. Table 1: maybe the authors should consider providing the p-values for those demographic variables so that the readers can tell whether the treatment and control group had similar baseline characteristics.

2. Multi-center trials (10 medical centers): has any analysis been conducted to evaluate the effect of different medical centers on the outcomes?

Reviewer #3 (Remarks to the Author):

Thank you for letting me review "a pilot randomized trial of Trendelenburg position for acute moderate ischemic stroke" by Hui-Sheng Chen to be published in Nature Communications.

The authors randomized patients with probable large artery atherosclerotic (LAA) stroke to either head down position (HDP) (-20 degree from 8:00AM-10:00 PM within the first 24hours after randomization plus 1-1.5 hours three times a day for the subsequent 10-14 days) or guideline recommended standard of care treatment without head-down position. LAA stroke probably translates to patients without atrial fibrillation (known before hospital admission or detected on admission ECG) as study inclusion had to occur before all diagnostic procedures would have been completed. Of note, patients who received intravenous thrombolysis (IVT) and/or endovascular therapy were not included (i.e. a population similar to the placebo groups in IVT trials).

96 patients were randomized. Endpoint was available in 46 (HDP) versus 48 (standard of care) patients (ITT). ITT showed neutral results for the primary endpoint (percentage of mRS=0-2 at day 90). ITT showed superiority of HDP for secondary endpoints (i.e. mRS=0-1 at day 90; shift analysis and change in NIHSS at day 12). Nearly four times as many patients achieved mRS=0 (no neurological deficits at all) at day 90 in the HDP group (23.9% versus 6.3%). Randomization occurred at 14.5hours (HDP) and at 10 hours (standard of care) after symptom onset on average. In terms of effect size, HDP had an OR of 2.05 for achieving mRS=0-1 (excellent outcom). This effect size of HDP is considerably better than intravenous thrombolysis given within 3-4.5 hours (ECASS 3, NEngJMed 2008, OR=1.34; ECASS population had a median NIHSS=9, too) although HDP was started later.

**There were no safety concerns with regard to HDP.
The paper is well written. No concerns with respect to language/grammar.**

The following aspects may help to further improve the paper:

Standard of care would include carotid intervention for extracranial symptomatic carotid stenosis (prevalent in 10/49 patients). Why was carotid intervention withheld? (Page 9, lines 210 ff.)

In the control group, only 25.1% achieved mRS=0-1 at 90 days. This is considerably less than the control group in ECASS 3 (NEngJMed 2008) with 45.1% achieving mRS=0-1 at 90 day (stroke severity NIHSS=9, too). Why did control patients perform so poorly in the study presented?

Introduction, page 5, lines 87 ff, Citation 18: The wording "moderate AIS-LAA" may be misleading, because "moderate" could relate to the degree of large artery atherosclerosis as well but refers to stroke severity. Consider rephrasing.

Please report ethics (IRB).

Side-lying positions with -20 degree was allowed if there was a high risk of aspiration suspected. How often/in how many patients did side-lying occur?

Sample size calculation : (Page 8, lines 175 ff) : The authors state, that "no relevant data [was available to perform sample size calculation] from previous trials". However, several reviews on HDP are cited. Why did previous trial data not fit? Please specify.

Discussion: Could disparity in responsible vessel have affected outcomes? Intracranial ICA stenoses (located proximally of the circle of Willis) seems to be more often present in the HDP group (21% vs 11%) whereas MCA-M1 stenoses (located distally to the circle of Willis) are less common in the HDP group (61%vs78%). Degree of collaterals may have played a role.

Many authors would consider the etiology in stroke recurrence in LAA stroke to be embolic in nature in the majority of patients (not hemodynamic) – why would HDP for 10-14 days affect embolic recurrence (until day 90)? (Stroke recurrence reported to be 2.3% (HDP) versus 8.5% at day 90).

The low aspiration rate is impressive. Only 1/95 patients developed stroke-associated pneumonia (SAP). Moreover, SAP occurred in the control group only. For comparison: Infection rate at day 11 was 15.4% (treatment group) compared to 32.5% (placebo group) in the Pantheris RCT (Meisel et al 2008). Please comment.

HDP was applied from 8:00 Am to 10:00PM during the first 24 hours after randomization. What happened, if the patient was randomized between 8:00AM -10:00 PM? Was the time period shortened?

Page 9 lines 211 ff. The sentence is very long. Consider rephrasing.

Safety and adverse events (page 10, lines 242 ff.): Please explain how "anxiety" and "fear" were differentiated and why table 4 reports "fear" only.

Table 4: Please define/clarify the outcome "tension".

References:

Some minor errors/typos. E.g. citation 13 is missing pages, citation 13 doubles the journal name, citation 29 includes additional information.

Dear Editor, dear Reviewers,

Re: Manuscript Title: A pilot randomized trial of Trendelenburg position for acute moderate ischemic stroke.

Thank you for your letter and for the reviewers' comments regarding our manuscript entitled "A pilot randomized trial of Trendelenburg position for acute moderate ischemic stroke". The comments are very helpful for revising and improving our paper. We have studied the comments carefully and revised the manuscript. The amendments have been tracked. Responses to the comments are listed below.

Response to the reviewers' comments:

Reviewer #1

Comment 1: Sample size of 94 patients seems to be too small, and this trial seems to be underpowered to show efficacy on the primary outcome of functional status at 90 days.

Reply: *Thank you for your important question. In the current literature, all the clinical trials about head position focused on the comparison of supine to sitting position in stroke, which is different from the comparison of -20 Trendelenburg position to supine or sitting position in the current trial. Given the nature of this phase 2 trial and no reference in the available literature, the sample size was set at 100 patients after discussion with the steering committee and finally 94 patients were enrolled. As you mention, the sample size was small and underpowered to demonstrate the efficacy on the primary outcome, which has been discussed as a limitation (Page 10, lines 224-226).*

Comment 2: Associated with insufficient sample size, there seems to be some

imbalance in important baseline factors, such as age, diabetes and systolic BP.

Reply: *Thank you for your valuable comment. Indeed, there seems to be some imbalance in important baseline factors, such as age, diabetes and systolic BP between groups due to the small sample size. We compared the baseline characteristics between groups in the control vs HDP group and found no significant difference between groups. We provided the p-values for those demographic variables and added this into the revised Table 1.*

Comment 3: The authors did not provide information on stroke management (ICU care, thrombolysis, endovascular treatment etc) which can affect primary outcome of physical function, although imbalance in stroke management can bias the results of this trial.

Reply: *Thank you for your professional advice. In the current trial, all patients who received reperfusion treatments such as thrombolysis and endovascular treatment were excluded, as we aimed to exclude the confounding factor of reperfusion treatment. This highly selected patient population may introduce selection bias and limit generalizability of our results, which has been discussed as a limitation (Page 10, line 228-231). Besides the head down position treatment in the HDP group, early management of ischemic stroke was based on current national stroke guidelines and followed in all patients, which was added in the revised manuscript (Page 12, lines 302-303). Only 19 patients received ICU care (11 in the HDP group and 8 in the control group), which was added to the revised Table 1.*

Reviewer #2

Comment 1: Table 1: maybe the authors should consider providing the p-values for those demographic variables so that the readers can tell whether the treatment and control group had similar baseline characteristics.

Reply: *Thank you for your professional advice. According to your suggestion, the p-values for the demographic variables have been added in the revised Table 1.*

Comment 2: Multi-center trials (10 medical centers): has any analysis been conducted to evaluate the effect of different medical centers on the outcomes?

Reply: *Thank you for this suggestion. Due to the nature of this phase 2 trial, the small size make it difficult to perform the analysis on the effect of different medical centers on the outcomes, which has been added as a limitation in the revised manuscript (Page 10, line 224-226).*

Reviewer #3

Comment 1: Standard of care would include carotid intervention for extracranial symptomatic carotid stenosis (prevalent in 10/49 patients). Why was carotid intervention withheld? (Page 9, lines 210 ff.)

Reply: *Thanks for your invaluable suggestion. Before enrollment, we assessed whether patients with LAA were eligible for carotid revascularization. For eligible patients, they were not enrolled if revascularization was planned within 90 days. Only patients who refused intervention within 90 days were included in the study.*

Comment 2: In the control group, only 25.1% achieved mRS=0-1 at 90 days. This is considerably less than the control group in ECASS 3 (NEngJMed 2008) with 45.1% achieving mRS=0-1 at 90 day (stroke severity NIHSS=9, too). Why did control patients perform so poorly in the study presented?

Reply: *Thank you for your question. The low proportion of mRS 0-1 at 90 days in this trial may be due to the highly selected patient population with large artery*

atherosclerosis who may have poorer outcome (Wang Y, Zhao X, Liu L, et al. Prevalence and outcomes of symptomatic intracranial large artery stenoses and occlusions in China: the Chinese Intracranial Atherosclerosis (CICAS) Study. Stroke 2014;45(3):663-669. doi:10.1161/STROKEAHA.113.003508).

Comment 3: Introduction, page 5, lines 87 ff, Citation 18: The wording “moderate AIS-LAA” may be misleading, because “moderate” could relate to the degree of large artery atherosclerosis as well but refers to stroke severity. Consider rephrasing.

Reply: *Thank you for your invaluable suggestion. We changed the wording “moderate AIS-LAA” according to the comment as “moderate AIS patients with LAA” and revised in the manuscript accordingly (Page 5, lines 92).*

Comment 3: Please report ethics (IRB).

Reply: The IRB has been added (Page 11, line 251-252).

Comment 4: Side-lying positions with -20 degree was allowed if there was a high risk of aspiration suspected. How often/in how many patients did side-lying occur?

Reply: *Thank you for your question. During the treatments, the side-lying position was used in 26.1% (12/46) patients to prevent possible aspiration. The related description was shown in the main text (Page 6, lines 112-113) and Table 2.*

Comment 5: Sample size calculation : (Page 8, lines 175 ff) : The authors state, that “no relevant data [was available to perform sample size calculation] from previous trials”. However, several reviews on HDP are cited. Why did previous trial data not fit? Please specify.

Reply: *Due to the absence of prior studies about the effect of -20 Trendelenburg position on stroke, we stated “no relevant data were available from previous studies”. In the current literature, all the clinical trials about the head position focused on the comparison of supine to sitting position in stroke. We may derive from a previous study ((Olavarría VV et al. Flat-head positioning increases cerebral blood flow in anterior circulation acute ischemic stroke. A cluster randomized phase IIb trial. Int J Stroke. 2018;13(6):600-611.) to calculate the sample. Their phase IIb study still consisted of the comparison of supine to sitting position in stroke, which is different from the comparison of -20 Trendelenburg position to supine or sitting position in the current trial. Thus, the sample calculation may not be accurate based on this study. Given that the current study is a phase 2 trial, we did not perform a formal sample size calculation after discussion with the steering committee.*

Comment 6: Discussion: Could disparity in responsible vessel have affected outcomes? Intracranial ICA stenoses (located proximally of the circle of Willis) seems to be more often present in the HDP group (21% vs 11%) whereas MCA-M1 stenoses (located distally to the circle of Willis) are less common in the HDP group (61% vs 78%). Degree of collaterals may have played a role.

Reply: *Thank you for your valuable advice. Indeed, location of responsible vessel may affect the outcome. According to your suggestion, we included locations of responsible vessels in the adjusted models as covariates and performed the analysis again. We found that the results remained unchanged (please see revised Table 3).*

Comment 7: Many authors would consider the etiology in stroke recurrence in LAA stroke to be embolic in nature in the majority of patients (not hemodynamic) – why would HDP for 10-14 days affect embolic recurrence (until day 90)? (Stroke recurrence reported to be 2.3% (HDP) versus 8.5% at day 90).

Reply: *Thank you for your question. As you mentioned, LAA pathogenesis includes arterial-to-arterial embolization and hypoperfusion. A recent study showed that hypoperfusion was more associated with recurrent stroke (Lyu J, Ma N, Tian C, et al. Perfusion and plaque evaluation to predict recurrent stroke in symptomatic middle cerebral artery stenosis. Stroke Vasc Neurol. 2019;4(3):129-134. doi:10.1136/svn-2018-000228). We contend that HDP decreased stroke recurrence due to improvement of hypoperfusion and in turn, may improve clearance of potential emboli from large artery atherosclerosis.*

Comment 8: The low aspiration rate is impressive. Only 1/95 patients developed stroke-associated pneumonia (SAP). Moreover, SAP occurred in the control group only. For comparison: Infection rate at day 11 was 15.4% (treatment group) compared to 32.5% (placebo group) in the Pantheris RCT (Meisel et al 2008). Please comment.

Reply: *Thank you for your astute comment. Indeed, the low aspiration rate was unexpected, which may be attributed to the following explanations. First, according to the inclusion/exclusion criteria, all patients had to complete the swallowing screen before enrollment. Patients were not enrolled if they had any possible contraindication to head-down position, such as active vomiting, swallowing disorders, or need for enteral feedings given a high risk of aspiration. Second, preventive measures such as side-lying were taken for patients with potential aspiration during treatment. Third, in the Pantheris RCT study, enrollment patients had severe stroke with NIHSS score 15-17, while the patients included in this study had moderate stroke with median NIHSS 9. Finally, in the Headpost RCT study (Olavarría VV et al. 2018) that compared supine with sitting position in stroke, the investigators found that the rate of pneumonia was 3.1% (supine group) compared to 3.4% (sitting group), with no significant between-group difference observed.*

Comment 9: HDP was applied from 8:00 Am to 10:00PM during the first 24

hours after randomization. What happened, if the patient was randomized between 8:00AM -10:00 PM? Was the time period shortened?

Reply: *Thank you for your valuable comment. The original purpose of setting this time period was to ensure that patients get enough sleep at night. In the course of treatment, it was adjusted according to the time and state of the patient. As you mentioned, the treatment time will be shortened if the patient was randomized between 8:00AM -10:00 PM. The median duration of position intervention within 24 hours was 15.0 h (IQR 12.0-16.0). The related description was shown in the main text (Page 6, lines 110-111) and Table 2.*

Comment 10: Page 9 lines 211 ff. The sentence is very long. Consider rephrasing.

Reply: *Thank you for your advice. According to your suggestion, we have deleted “, except for a higher prevalence of risk factors such as diabetes, coronary heart disease, current drinker and smoker, higher rate of ICA occlusion in the HDP group vs control group, and less hypertension and older age in the control vs HDP group”, because there was no significance although numerically higher or less in the control vs HDP group (Page 6, lines 105-108).*

Comment 11: Safety and adverse events (page 10, lines 242 ff.): Please explain how “anxiety” and “fear” were differentiated and why table 4 reports “fear” only.

Reply: *Sorry for the error. The “tension” in table 4 should be “anxiety”. The character of anxiety symptoms included exhaustion, worry and being depressed, etc. The fear shows a sudden strong feeling of terror, similar to a panic attack.*

Comment 12: Table 4: Please define/clarify the outcome “tension”.

Reply: *As replied above, the “tension” in table 4 should be replaced with “anxiety”.*

Comment 13: References: **Some minor errors/typos. E.g. citation 13 is missing pages, citation 13 doubles the journal name, citation 29 includes additional information.**

Reply: *Sorry for this oversight. According to your suggestion, the corresponding reference information was modified in the revised manuscript.*

REVIEWER COMMENTS

Reviewer #1 (Remarks to the Author):

Thank you for nice revision of the paper. I still have some more comments.

1. Thank you for adding information on ICU care but more detailed information is required (e.g. use of rtPA, endovascular treatment, drugs used for acute management, BP levels during acute phase etc).
2. Even if the differences in clinical features are not statistically significant, there are clinically important differences (e.g. age 61 vs 64, ICA stenosis 21% vs 11%, onset to randomization 15h vs 10h etc). These differences might have affected the results of this paper. The authors should conduct sensitivity analyses with adjustment for these factors.
3. There might be imbalance in stroke management. These factors should also be adjusted in sensitivity analyses.

Reviewer #2 (Remarks to the Author):

My concerns have been addressed in the revised manuscript.

Reviewer #3 (Remarks to the Author):

The authors have eloquently addressed the raised questions.

Dear Editor, dear Reviewers,

Re: NCOMMS-22-49632B, entitled “A pilot randomized trial of Trendelenburg position for acute moderate ischemic stroke”

Thank you for your letter and for the reviewers’ comments regarding our manuscript entitled “A pilot randomized trial of Trendelenburg position for acute moderate ischemic stroke”. The comments are very helpful for revising and improving our paper. We have studied the comments carefully and revised the manuscript. The amendments have been tracked. Responses to the comments are listed below.

Response to the reviewers’ comments:

Reviewer #1

Comment 1: Thank you for adding information on ICU care but more detailed information is required (e.g. use of rtPA, endovascular treatment, drugs used for acute management, BP levels during acute phase etc).

Reply: *Thank you for your professional advice. In order to exclude the confounding factor of reperfusion treatment, patients who received thrombolysis and endovascular treatment were excluded in this trial. Besides the head down position treatment in the HDP group, early management of ischemic stroke was based on current national stroke guidelines. According to your suggestion, we supplemented the information of drugs used for acute management such as lipid-lowering therapy, and blood pressure levels during the acute phase, which was added to the revised Table 1.*

Comment 2: Even if the differences in clinical features are not statistically significant, there are clinically important differences (e.g. age 61 vs 64, ICA stenosis 21% vs 11%, onset to randomization 15h vs 10h etc). These differences

might have affected the results of this paper. The authors should conduct sensitivity analyses with adjustment for these factors.

Reply: *Thank you for your valuable advice. In the original manuscript, we have included age and location of responsible vessels as covariables in the sensitivity analyses. According to your suggestion, we included onset to randomization in the adjusted models as covariates and performed the analysis again. We found that the results remained unchanged (please see Table 3).*

Comment 3: There might be imbalance in stroke management. These factors should also be adjusted in sensitivity analyses.

Reply: *Thank you for your invaluable comment. For stroke management, we supplemented and analyzed the antithrombotic therapy, lipid-lowering therapy and blood pressure levels during the acute phase, and found no significant difference between groups, thus we did not perform sensitivity analysis with adjustment for these variables. We provided the p-values for those variables in the revised Table 1.*

REVIEWERS' COMMENTS

Reviewer #1 (Remarks to the Author):

Thank you for nice revision of the paper.